# Immunotherapy for Colorectal Cancer with High Microsatellite Instability: The Ongoing Search for Biomarkers

**DOI:** 10.3390/cancers15174245

**Published:** 2023-08-24

**Authors:** Javier Ros, Iosune Baraibar, Nadia Saoudi, Marta Rodriguez, Francesc Salvà, Josep Tabernero, Elena Élez

**Affiliations:** 1Medical Oncology Department, Vall d’Hebron University Hospital, 08035 Barcelona, Spain; fjros@vhio.net (J.R.); ibaraibar@vhio.net (I.B.); nsaoudi@vhio.net (N.S.); martarodriguez@vhio.net (M.R.); fsalva@vhio.net (F.S.); jtabernero@vhio.net (J.T.); 2Vall d’Hebron Institute of Oncology, 08035 Barcelona, Spain

**Keywords:** MSI colorectal cancer, immunotherapy, biomarkers, liver metastases, adjuvant, neoadjuvant

## Abstract

**Simple Summary:**

Immunotherapy has reshaped the prognosis of several tumor types. In metastatic colorectal cancer, only tumors with microsatellite instability or mismatch repair deficiency achieve profound benefits under immune checkpoint inhibitors. As a result, immunotherapy has moved to the early stage, showing promising results in neoadjuvant settings. However, not all patients respond, and some responses are shortlived, thus highlighting the need for accurate and reliable biomarkers to identify patients who are more likely to achieve clinical benefit, as well as those patients with refractory tumors who will require a different therapeutic approach. Surprisingly, classical biomarkers such as PD-L1 expression or TMB seem to be poorly informative in MSI colorectal cancer. Therefore, the development of novel biomarkers in this population remains an unmet medical need.

**Abstract:**

Microsatellite instability (MSI) is a biological condition associated with inflamed tumors, high tumor mutational burden (TMB), and responses to immune checkpoint inhibitors. In colorectal cancer (CRC), MSI tumors are found in 5% of patients in the metastatic setting and 15% in early-stage disease. Following the impressive clinical activity of immune checkpoint inhibitors in the metastatic setting, associated with deep and long-lasting responses, the development of immune checkpoint inhibitors has expanded to early-stage disease. Several phase II trials have demonstrated a high rate of pathological complete responses, with some patients even spared from surgery. However, in both settings, not all patients respond and some responses are short, emphasizing the importance of the ongoing search for accurate biomarkers. While various biomarkers of response have been evaluated in the context of MSI CRC, including *B2M* and *JAK1*/2 mutations, TMB, *WNT* pathway mutations, and Lynch syndrome, with mixed results, liver metastases have been associated with a lack of activity in such strategies. To improve patient selection and treatment outcomes, further research is required to identify additional biomarkers and refine existing ones. This will allow for the development of personalized treatment approaches and the integration of novel therapeutic strategies for MSI CRC patients with liver metastases.

## 1. Introduction

Colorectal cancer (CRC) is a prevalent cancer type, ranking third worldwide in frequency of diagnosis and second in cancer-related deaths, as reported by Globocan estimates [1]. Despite advances in early detection and novel treatments, the five-year survival rate for CRC patients in the USA remains low, at 12.5% [2]. However, it is widely acknowledged that the specific molecular profile of a given tumor may substantially change the prognosis. Indeed, in the metastatic setting, mutational status and genomic characteristics have deep prognostic and predictive repercussions. While RAS/BRAF wild-type tumors are associated with longer overall survival (OS), BRAF-V600E-mutant tumors have a very poor prognosis [3,4]. The development of targeted agents has improved the prognosis of several CRC molecular subgroups; however, not all patients respond, and some responses are relatively shortlived. In-depth knowledge of the underlying biology of metastatic CRC (mCRC) has allowed for better stratification when selecting therapies. In this context, microsatellite instability (MSI) status plays a distinct role, as this genomic condition has proven not only to be a prognostic factor but also to be predictive of response to therapeutic strategies using immunomodulation.

In recent years, immune checkpoint inhibitors have transformed the prognosis of certain tumor types, including melanoma, as well as renal, bladder, and lung cancers, achieving deep, durable, and even complete responses. However, identifying biomarkers of response is critical to identifying patients who can benefit from immunotherapy. In the case of mCRC, immune checkpoint inhibitors have shown impressive and durable responses in tumors with high MSI (MSI-h) or mismatch repair deficiency (dMMR); however, these account for only 5% of all mCRC cases [5], with the vast majority of mCRC being microsatellite stable (MSS). Unfortunately, among MSS tumors, immune-based strategies have demonstrated poor clinical activity [6]. Nonetheless, it is noteworthy that the prevalence of MSI in early-stage CRC is higher, at approximately 15% of cases [7]. The role of immune checkpoint inhibitors in stage II–III disease is under investigation.

Microsatellites are repetitive DNA sequences that do not encode for proteins, with a length ranging from one to ten nucleotide base pairs. Due to their high susceptibility to mutation during DNA replication, they serve as common locations for DNA errors. The mismatch repair system encompasses four proteins (MLH1, PMS2, MSH2, and MSH6) responsible for detecting and correcting these errors, by acting as functional heterodimers (MLH1/PMS2 and MSH2/MSH6).

Sporadic cases of MSI are caused by the inactivation of *MMR* genes through somatic mutations or epigenetic silencing (e.g., *MLH1* promoter somatic hypermethylation), frequently associated with *BRAF* mutations. MSI-high is a hallmark of Lynch syndrome, also known as hereditary nonpolyposis CRC (HNPCC), an autosomal dominant genetic syndrome characterized by an increased risk of developing certain types of cancers, particularly CRC and endometrial cancers. The ESMO guidelines for CRC recommend testing for MMR status at an early stage—stage II—at which time the presence of MSI is associated with a good prognosis, therefore avoiding adjuvant chemotherapy in some cases. MSI status should be systematically determined upfront in the metastatic setting wherever possible, in light of the therapeutic implications [8,9,10,11]. There are three different techniques to determine MSI status: immunohistochemical staining, polymerase chain reaction (PCR), and next-generation sequencing [8]. While immunohistochemistry is a cost-effective and widely accessible tool, it does not capture rare protein-function-altering mutations. PCR-based MSI detection necessitates a tumor sample with an adequate number of tumor cells for DNA extraction. Although neither method is 100% sensitive, both exhibit a high level of concordance in CRC (90–97%). PCR-based MSI testing, while more demanding in terms of tissue quality and quantity, and more costly, offers a reliable and well-established approach in CRC. It employs an MSI-PCR panel of markers—typically a set of five mononucleotides known as the pentaplex panel in Europe (including NR-27, NR-21, NR-24, BAT-25, and BAT-26), as recommended by the National Cancer Institute. Rather than targeting specific harmful mutations in MMR genes, this method identifies the genomic effects caused by the loss of MMR protein function. The MSI phenotype is determined by the presence of at least three unstable markers out of the five analyzed (or two when compared to healthy tissue) [12,13].

MSI-H/dMMR tumors have a high accumulation rate of mutations, leading to the formation of frameshift proteins, or neoantigens, which can be recognized by the immune system, along with high levels of tumor-infiltrating lymphocytes (TILs). These tumors commonly occur in the proximal colon, have poor differentiation, display mucinous histology, and are enriched in tumors harboring BRAF-V600E mutations [14,15,16,17]. A number of well-defined tumoral biological features have been identified that may explain the clinical behavior of both MSI and MSS tumors. Immunosuppressive cells are diminished and immunoinhibitory molecules are elevated in hypermutated tumors. Conversely, non-hypermutated tumors have an abundance of immunosuppressive cells, with downregulation of the expression of immune inhibitors and MHC molecules [18]. Indeed, MSI-H/dMMR tumors are often infiltrated by immune cells (such as CD4+, CD8+, Th1, and macrophages), display upregulated expression of immune checkpoint proteins (such as PD-1, PD-L1, and CTLA4), and often have low frequency of myeloid-derived suppressor cells (MDSCs) and regulatory T cells (Tregs). These features suggest that MSI-H/dMMR CRCs might have a favorable response to immune checkpoint inhibitors [19,20,21]. Conversely, immune checkpoint inhibitors, alone or in combination with other drugs (such as tyrosine kinase or MEK inhibitors, anti-EGFR, or chemotherapy), have demonstrated poor activity in the majority of tumors that are mismatch-repair-proficient or MSS [6].

Several clinical trials have demonstrated deep and durable responses, including complete responses, in both metastatic and neoadjuvant settings when immune checkpoint inhibitors are used in MSI tumors. However, with some MSI patients not responding to immunotherapy strategies, while in certain responders the responses are not long-lasting, this highlights a critical need to better identify the mechanisms of resistance and gain a deeper understanding of the interaction between immune checkpoint inhibitors and the tumor environment. Here, we review all published clinical trials with immune checkpoint inhibitors in MSI tumors and other emerging immune strategies, and we explore the outcomes in different disease settings. We also summarize current research into the identification of biomarkers for potential mechanisms of resistance, and we assess how this may be exploited to optimize the treatment of patients with MSI CRC. Table 1 summarizes the outcomes of all completed clinical trials reported to date evaluating various immune checkpoint inhibitors in MSI mCRC.

## 2. Development of Immune Therapeutics in MSI mCRC

### 2.1. Immune Checkpoint Inhibitors in the Advanced/Refractory Setting

The development of immune checkpoint inhibitor therapies in MSI CRC has revolutionized the treatment landscape, offering a promising avenue to harness the patient’s own immune system against the tumor. Several trials have evaluated the clinical activity and safety of immunotherapy in MSI colorectal cancer, and the benefit in terms of survival has been confirmed in a meta-analysis [33]. A range of anti-PD-1, anti-PD-L1, and anti-CTLA-4 agents have been evaluated as single agents or in combination in the second- and later-line settings, and more recently in the first-line setting. Outcomes and biomarker analyses are discussed. Figure 1 summarizes the mechanism of action of immune checkpoint inhibitors.

The Keynote-016 phase II trial investigated the efficacy and safety of pembrolizumab, an anti-PD-1 antibody, in patients with refractory colorectal tumors classified as either MSI (n: 11) or MSS (n: 21). This trial not only revealed the limited effectiveness of immune checkpoint inhibitors in MSS mCRC but also demonstrated remarkable and enduring responses in the MSI subpopulation, with an overall response rate (ORR) of 40% [24]. Additionally, correlative analysis indicated that MSI tumors exhibited higher levels of TILs and higher expression of PD-L1 [24], although this was not associated with better outcomes.

Pembrolizumab was assessed in the refractory setting through the Keynote-164 phase II trial [22]. This trial comprised two cohorts: cohort A, which enrolled 61 patients with two or more previous lines of treatment, and cohort B, which included 63 patients with one or more previous lines of treatment. After a median follow-up of 62 and 54 months, respectively, cohort A demonstrated an ORR of 32.8%, while cohort B exhibited an ORR of 34.9%. The median PFS was reported as 2.3 months for cohort A and 4.1 months for cohort B, whereas the median OS was 31 months for cohort A and 47 months for cohort B. Notably, complete responses were observed in 4.9% of patients in cohort A and 14.3% of patients in cohort B. These findings align with the observations from the CheckMate-142 trial, suggesting that immune checkpoint inhibitors may exhibit greater efficacy in earlier lines of treatment compared to the refractory setting.

Subsequently, several phase I or I/II studies assessed the clinical efficacy and safety of the anti-PD-1 or anti-PD-L1 agents durvalumab, avelumab, and dostarlimab in MSI mCRC, also in the refractory setting, and also showing encouraging efficacy outcomes. ORRs ranged from 22% to 36%, while median progression-free survival (PFS) durations ranged between 3.9 and 5.5 months. The proportion of non-responders, characterized by progressive disease as the best overall response, ranged from 18% to 30% [29,30,32,34]. Avelumab was also evaluated in the second-line setting in the single-arm, phase II SAMCO-PRODIGE 54 trial, which included 132 patients. The ORR was 29.5%, with a median PFS of 3.9 months and median OS not reached after 33 months of follow-up [31].

Combination therapy with nivolumab (anti-PD-1) and ipilimumab (anti-CTLA-4) was evaluated in the refractory setting in the NIPICOL and CheckMate-142 trials. The NIPICOL trial enrolled 57 patients with MSI mCRC and investigated the efficacy of nivolumab plus ipilimumab. The three-year PFS and OS rates were found to be 70% and 73%, respectively, highlighting the impressive clinical activity achieved with this dual-blockade approach.

CheckMate-142 was a large phase II trial with three cohorts: nivolumab plus ipilimumab in the first-line setting, nivolumab plus ipilimumab in the refractory setting, and nivolumab monotherapy in the refractory setting. It is important to note that this was not a randomized trial, and its design did not therefore allow for direct comparisons between the three arms. The nivolumab monotherapy arm included 74 patients and demonstrated an ORR of 34%, with 9% achieving complete responses and 30% classified as non-responders. After a median follow-up of 21 months, the median PFS was 6.6 months, while the median OS had not been reached. The nivolumab–ipilimumab refractory cohort, which enrolled 119 patients, demonstrated an ORR of 65%, with 13% of the patients achieving complete responses and 12% classified as non-responders. Interestingly, earlier reports with shorter follow-up described lower ORRs (55% and 58% after 13.4 and 25.4 months of follow-up, respectively), suggesting that the longer the follow-up, the higher the response rate. Likewise, complete response rates increased with longer follow-up, rising from 3% after 13.4 months to 13% after 50 months of follow-up, suggesting an increase in both the rate and depth of responses over time. The median PFS and OS had not been reached after a median follow-up of 50 months [26].

The CheckMate-142 cohort evaluating nivolumab plus ipilimumab in the first-line setting included 45 patients and demonstrated an ORR of 69%, with 13% achieving complete responses and 13% classified as non-responders. Similar to the previous cohorts, the median PFS and OS had not been reached after 29 months of follow-up [27]. While it is important to note that this trial was not randomized, making formal comparisons difficult, the results of CheckMate-142 showcase the superiority of the double combination over nivolumab monotherapy and suggest that earlier administration of immune checkpoint inhibitors may yield better outcomes. Moreover, unlike other therapeutic strategies such as chemotherapy, the responses appeared to increase with longer follow-up. Subgroup analysis also demonstrated the superiority of nivolumab–ipilimumab across all subgroups, regardless of age, sex, ECOG performance status, or mutational status. However, the evaluation of other biomarkers, such as PD-L1 expression, yielded inconclusive results. In the nivolumab cohort, patients with high PD-L1 (>1%) expression exhibited similar ORRs but poorer 12-week disease control rates compared to patients with low PD-L1 expression (28% vs. 27% and 52% vs. 74%, respectively). In the nivolumab–ipilimumab refractory cohort, high PD-L1 expression was associated with a higher ORR (70% vs. 61%).

Finally, the Keynote-177 trial stands as the only randomized, phase III clinical trial investigating immune checkpoint inhibitors in mCRC [23]. This study compared pembrolizumab with chemotherapy in the first-line treatment setting. The primary objectives of the trial were based on two co-criteria: PFS and OS. With a median follow-up of 32.4 months, pembrolizumab demonstrated a significant improvement in median PFS compared to chemotherapy (16.5 vs. 8.2 months), effectively doubling the PFS. Although pembrolizumab showed a superior OS, statistical significance was not achieved, potentially due to a crossover rate of 60% in the trial. Nonetheless, 30% of patients treated with pembrolizumab were non-responders, in contrast to 12% of patients treated with chemotherapy. Several factors may have contributed to this high rate of non-responders, including intrinsic resistance to treatment, misdiagnosis of MSI status (possibly resulting from errors in immunohistochemistry techniques), and pseudoprogression events.

### 2.2. Other Immune Strategies in the Advanced/Refractory Setting

Adoptive cell transfer (ACT) is an immunotherapeutic approach that targets tumor-specific antigens presented by MHC proteins and recognized by T cells, triggering an antitumor T-cell immune response. Neoantigens, which are unique to the tumor and arise from somatic mutations, are identified using NGS technologies. ACT can involve the manipulation of TILs or the host’s T cells that have been genetically altered to express a T-cell receptor or a chimeric antigen receptor (CAR) [35,36,37]. However, neoantigens are rarely shared between patients, highlighting that cancer vaccination requires an individualized strategy [18].

ACT therapy has shown clinical responses in cholangiocarcinoma, breast cancer, metastatic melanoma, and CRC [38,39,40]. In a phase II clinical trial assessing the efficacy of adoptive transfer of autologous TILs, one patient with mCRC showed objective regression; following one infusion with TILs reactive to a *KRAS* G12D mutation identified in the patient’s tumor, the patient presented with regression of all seven lung metastases [41].

CAR T-cell therapy has also been explored in the setting of CRC. CAR T cells can be manipulated to target tumor-associated antigens that are highly expressed by CRC tumors, such as CEA. A phase I clinical trial indicated that CEA CAR T-cell therapy showed some efficacy in mCRC patients with CEA-positive tumors (7 out of 10 patients achieved stable disease), with an acceptable toxicity profile [42]. Another phase I trial addressed the efficacy of CAR T cells targeting CEA as a local (intra-arterial) treatment for liver metastases in CRC [43], and signs of efficacy were reported. Neoantigen-targeting strategies have the potential to overcome the toxicities and have narrowed the response rates of non-antigen-specific treatments, making ACT a promising approach to improve the immune response in CRC patients [44].

Several tumor vaccines have been studied in mCRC, including peptide vaccines, autologous vaccines, dendritic cell transplants, and oncolytic viral vector vaccines [45,46]. The use of viral antigen vaccines is based on their ability to generate a strong immune response. Additionally, peptide-vaccine-based immunotherapy targets tumor-associated antigens that are overexpressed on the surface of tumor cells, such as CEA, melanoma-associated antigen, and MUC1. A phase II trial showed improved survival in mCRC patients who received autologous dendritic cells modified with a pox vector encoding MUC1 and CEA [47]. However, evidence for reliable survival benefits from cancer vaccines is still limited.

## 3. Immune Checkpoint Inhibitors for MSI CRC in Early-Stage Disease

### 3.1. Immune Checkpoint Inhibitors in the Adjuvant Setting

Currently, there are three clinical trials with immune checkpoint inhibitors in the adjuvant setting for MSI CRC, all of which are ongoing. ATOMIC (NCT02912555) is a phase III trial evaluating the combination of adjuvant FOLFOX chemotherapy with atezolizumab versus FOLFOX alone, with the primary endpoint of disease-free progression. The trial design specifically excludes an atezolizumab monotherapy arm. The results of this trial are yet to be published. The POLEM trial (NCT03827044) is comparing adjuvant 5FU-based chemotherapy followed by avelumab to adjuvant 5FU-based chemotherapy alone in patients with MSI or POLE mutations and stage III colon cancer. The primary endpoint is 3-year disease-free survival. As with the ATOMIC trial, the results of the POLEM trial have not yet been published. Finally, the anti-PD-1 tislelizumab is being investigated in a single-arm phase II trial, as adjuvant monotherapy for patients with high-risk stage II and III dMMR colon cancer (NCT05231850).

### 3.2. Neoadjuvant Setting

Evidence from several tumor types suggests that neoadjuvant immune checkpoint inhibitors may lead to a deep pathological response [48,49,50,51]. The outcomes of immune-enhancing strategies in early-stage cancer are thought to be associated with lower tumor burden, higher TILs, and a lower degree of systemic immunosuppression. Based on these observations, neoadjuvant immune checkpoint inhibitors are now being tested in early-stage CRC. Table 2 summarizes the clinical trials evaluating immune checkpoint inhibitors as neoadjuvant therapy in MSI CRC.

The NICHE trial evaluated two doses of nivolumab and a single dose of ipilimumab, with or without the COX-1 inhibitor celecoxib, as a neoadjuvant therapy for early-stage MSI and MSS CRC [52]. Celecoxib was added (only in patients with MSS tumors) based on preclinical data suggesting that prostaglandin E2 may lead to subversion of myeloid cells and increase tumor-promoting inflammation [57]. Among MSI tumors, 65% of patients achieved a complete pathological response and 95% achieved a major pathological response. Following the paramount results of the NICHE trial, the NICHE-2 phase II study included 112 patients with MSI tumors. Patients were treated with one dose of ipilimumab and two doses of nivolumab and underwent surgery within 6 weeks [53]. A major pathological response rate of 95%, including 67% pathological complete responses, was reported.

The PICC trial was a phase II randomized Chinese trial in which 34 patients with localized CRC were randomized to the anti-PD-1 drug toripalimab, with or without celecoxib [54]. Pathological complete responses were reported in 65% and 88% of patients for toripalimab alone and toripalimab with celecoxib, respectively, and major pathological responses were reported in 94% and 100%, respectively; no relapses were reported after a median follow-up of 14 months.

Similarly, a phase II trial (NCT04165772) evaluated the potential role of the anti-PD-1 drug dostarlimab in patients with rectal cancer [55]. Only 12 patients were included; however, all of them achieved a complete pathological response, and after a median follow-up of 12 months there was no evidence of their tumors on magnetic resonance imaging, PET-CT, endoscopic evaluation, digital rectal examination, or biopsy. All patients were spared chemoradiotherapy and surgery, which highlights the outstanding effect of immunotherapy in the early stage, preserving patients’ quality of life. Regarding biomarkers, serial biopsies showed an initial expansion of PD-L1 and CD8 T cells, followed by a decrease at the time when the complete response was achieved. Taken together, dostarlimab alone has achieved significantly greater clinical benefit than that achieved with a conventional total neoadjuvant approach for patients with MSS rectal cancer [58,59].

Finally, the MDACC phase II trial evaluated the activity of neoadjuvant pembrolizumab in a cohort of patients with gastrointestinal malignancies, including 27 patients with CRC [56]. Among 14 evaluable CRC patients, the pathological complete response was 79%. It should be noted that four patients demonstrated clinical, radiological, or ctDNA progression; among them, three had N2 stage, two had the V600E *BRAF* mutation, and all four had a positive ctDNA baseline. Only one of them achieved a partial response. Pathological response analysis was only available for two patients, neither of whom achieved a pathological response. Unlike the dostarlimab trial, complete radiological responses were infrequent (i.e., 2/11 patients with pathological complete response at resection). Biomarker analysis suggested that a higher abundance of CD15 granulocytic cell types was associated with loss of treatment response. Overall, all of these trials support the value of the activity of neoadjuvant immune checkpoint inhibitors in both colon and rectal early-stage carcinomas, and they confirm that strategies with immune checkpoint inhibitors are safe, feasible, and result in a high rate of complete pathological response, highlighting the importance of early identification of MSI cancers. Furthermore, a meta-analysis including 410 cases of non-metastatic colorectal cancer (113 MSI tumors, 167 MSS tumors) treated with neoadjuvant immunotherapy confirmed that among those patients with MSI tumors, neoadjuvant treatment with immune checkpoint inhibitors achieved a greater rate of pathological complete response and major pathological responses [60].

## 4. Mechanisms of Resistance and Associated Biomarkers

Most patients with MSI mCRC treated with immune checkpoint inhibitors achieve meaningful clinical benefit, including approximately 10% of patients who achieve a complete response [22,23,26]. However, progression is observed in 10–45% of these patients, representing an important need to identify biomarkers of response that may help to tailor treatment, assess prognosis, and improve the clinical efficacy of immune checkpoint inhibitors. Figure 2 summarizes potential predictive biomarkers evaluated in MSI CRC.

While several mechanisms of resistance have been well characterized in other tumor types, in the case of MSI mCRC there is a lack of well-established biomarkers [61,62]. Indeed, acquired resistance to immunotherapy in patients with solid tumors has been associated with defects in the pathways involved in interferon-receptor signaling and antigen presentation. However, in CRC, these biomarkers seem to be poorly predictive.

### 4.1. Antigen Presentation Defects: B2M and JAK1/2

The *B2M* gene encodes the protein β2-microglobulin, an extracellular component of major histocompatibility complex (MHC) class I molecules that is present on all nucleated cells. MHC class I molecules have immune system self-recognition functions. Acquired *B2M* mutations and loss of B2M expression have been implicated as causes of acquired resistance to immunotherapy in melanoma [61,63]. *B2M* mutations occur in 13–24% of MSI-H CRCs and are usually associated with loss of B2M expression [64]. The exact predictive role of B2M in CRC remains controversial. Unlike other tumor types in which *B2M* mutations have been associated with a lack of response to immune checkpoint inhibitors, among immunotherapy-naïve CRC patients, up to 85% of tumors with *B2M* mutations achieve clinical benefit with immunotherapy, most notably in MSI-H tumors [65].

In addition, Janus kinase 1 and 2 (*JAK1*/2) mutations have also been proposed as genetic mechanisms of immune evasion for anti-PD1 therapy. In a small cohort including 35 patients with MSI mCRC treated with an anti-PD-1 drug, 29% had a *B2M* loss-of-function mutation, and 23% had a *JAK1*/2 loss-of-function mutation. Compared with *B2M* wild-type CRC, *B2M*-mutated CRC achieved better ORRs (70% vs. 64%) with anti-PD-1 therapy. Furthermore, there was better response to anti-PD-1 therapy in patients with *JAK1*/2 mutations than in those without them (ORR 100% vs. 55%). Similarly, among 110 MSI mCRC patients from the cBioPortal MSKCC cohort, taking *B2M* and *JAK1*/2 status together, patients with a *B2M* or *JAK1*/2 mutation had better OS compared with wild-type cases [66,67]. Overall, these results suggest that, unlike other tumor types, MSI CRCs with *B2M* and *JAK1*/2 mutations are responsive to PD-1 inhibitors, indicating that the mechanism of resistance to anti-PD-1 therapy in CRC may be different from that in other solid tumors [68]. Finally, because dMMR cancers that have genetic inactivation of β2M continue to show positive responses to immune checkpoint blockade, it is thought that there may be other immune effector cells involved in these responses other than CD8 T cells. A correlation between the inactivation of B2M and heightened infiltration of γδ T cells in MSI CRC has recently been reported. Using paired tissue samples from patients who received immune checkpoint inhibitors, an increase in the presence of γδ T cells in *B2M*-deficient cancers was observed, suggesting that γδ T cells play a role in the response to immune checkpoint blockade in patients with dMMR colon cancers; this could be exploited in future therapeutic strategies [69].

### 4.2. Impact of RAS/BRAF and WNT/β-Catenin Mutations in Colorectal Cancer

*MAPK* mutations have been associated with worse prognosis compared to patients with *RAS/BRAF* wild-type CRC [4,70,71]. However, data from phase II trials showed similar ORRs among MSI-H/dMMR patients with *RAS/BRAF*-mutated tumors compared to wild-type tumors treated with immune checkpoint inhibitors, and these same studies, along with independent cohorts, have demonstrated that *RAS/BRAF* mutations have no impact in terms of PFS in patients treated with immune checkpoint inhibitors [26,72,73,74]. Independent cohorts have also suggested that *RAS/BRAF* mutations do not decrease OS in patients with MSI mCRC treated with immunotherapy [65]. Nevertheless, the post hoc subgroup analysis of the Keynote-177 trial showed that mCRC patients with *RAS*-mutant tumors have worse PFS, albeit approximately 30% of these patients did not have data on mutational status available [75]. Truncating mutations in WNT/β-catenin have been associated with immunosuppression and, therefore, with lower clinical efficacy of immune checkpoint inhibitors. In CRC, higher levels of activation of the WNT pathway have been associated with lower TMB, leading to immune-cold tumors [76]. However, these findings have not been prospectively validated in clinical cohorts.

### 4.3. Lynch Syndrome

Lynch-associated CRCs exhibit a unique tumorigenesis pathway and distinct clinicopathological features compared to sporadic tumors [77,78,79]. Indeed, Lynch-associated CRC and endometrial cancers typically display more prominent local T-cell infiltration and have a higher mutational burden in comparison to sporadic MSI-H CRC, suggesting that there might be distinct responses to immune checkpoint inhibitors between these groups [80,81]. Moreover, in two prospective cohorts where the presence of Lynch syndrome was prospectively evaluated, patients with Lynch syndrome CRC treated with immune checkpoint inhibitors exhibited significantly better PFS and a trend towards better OS compared to sporadic cases [74]. This finding has also been reported in the neoadjuvant setting, in the NICHE-2 trial, in which patients with Lynch-syndrome-localized colon cancer treated with neoadjuvant ipilimumab plus nivolumab achieved a higher pathological complete response rate compared to sporadic tumors (78% vs. 58%) [52].

### 4.4. Tumor Mutational Load as a Predictive and Prognostic Biomarker

In several tumor types, pembrolizumab has been approved by the FDA for patients who have treatment-refractory cancers with a TMB of at least 10 mutations per megabase (Mut/Mb), based on their having higher response rates compared to patients with a lower TMB [82,83]. However, in CRC, TMB should be considered based on the mismatch repair status. Indeed, only MSI tumors achieve benefit with immune checkpoint inhibitor strategies, regardless of TMB, in contrast to what is observed in MSS tumors, where TMB seems to be non-informative [84]. However, the exact TMB cutoff has not been established [85]. In a cohort of 22 patients treated with PD-1/L1 inhibitors, TMB showed the strongest association with objective response and PFS. The optimal predictive cutoff point for TMB was estimated between 37 and 41 Mut/Mb. All patients with high TMB responded, while 65% of patients with low TMB had progressive disease. The median PFS for high TMB was not reached after a median follow-up of 18 months, while the median PFS for low TMB was 2 months [86]. Recently, the role of TMB in MSI mCRC treated with immunotherapy was addressed in a cohort of 110 patients treated with immune checkpoint inhibitors. The optimal prognostic cutoff for PFS in this cohort was 23 Mut/Mb. Patients with TMB ≤ 23 Mut/Mb had significantly worse PFS and OS. Patients with dMMR/MSI-H mCRC and relatively lower TMB values displayed early disease progression when receiving immune checkpoint inhibitors, whereas patients with the highest TMB values obtained the maximal benefit from intensified anti-CTLA-4/PD-1 combination therapy [87]. Based on these (albeit minimal) data, TMB may be insufficient to predict response in CRC [76], and further data from prospective, larger cohorts are needed to better elucidate the predictive value of TMB in MSI mCRC.

### 4.5. PD-L1 Expression

The results of the nivolumab-plus-ipilimumab cohort in the CheckMate-142 study gave an ORR of 54% both in tumors with PD-L1 expression ≥ 1% and < 1%. Disease control for more than 4 months was reported in 77% and 78% of patients, respectively, suggesting that PD-L1 expression based on this cutoff is not a reliable biomarker in CRC [20]. Similarly, in the phase II trial of pembrolizumab, PD-L1 expression (>5% vs. <5%) showed no differences in terms of ORR, PFS, or OS [24]. Currently, the EudraCT 2021-001309-60 trial is evaluating immunotherapy vs. the standard of care in previously treated metastatic PD-L1-positive CRC (CPS ≥ 1).

### 4.6. Liver and Adrenal Metastases

The presence of liver metastases has been associated with a lack of activity of immune checkpoint inhibitors, regardless of MMR status. Indeed, in the phase I trial evaluating regorafenib in combination with nivolumab and ipilimumab in MSS refractory mCRC, patients without liver metastases achieved an ORR of 40%, with a median PFS of 5 months, whereas among patients with liver metastases the ORR was 0% and the PFS was 2 months [88]. Similarly, in the phase I C-800 trial that evaluated the combination of botensilimab—a novel anti-CTLA4 agent—with the anti-PD-1 agent balstilimab, a higher response rate was seen in patients without liver metastases (42% vs. 24%) [89]. In MSI tumors, the same tendency has been observed. In a cohort of patients with MSI CRC treated with pembrolizumab in the first line, patients without liver metastases had a 63% ORR and 34-month median PFS, compared to 21% and 6 months, respectively, in patients with liver metastases [90]. These results suggest that liver metastases may induce antitumor immunity inhibition and immune tolerance, leading to T-cell exclusion. Liver metastases are enriched in TGF-β, where CD4 and CD8 T cells exhibit decreased expression, resulting in the disruption of immune tolerance. This is facilitated by the activation of liver type 1 macrophages and an increased presence of regulatory T cells and myeloid-derived suppressor cells [6,91,92,93]. Finally, a small cohort of patients suggested that adrenal gland metastases may also act as sanctuary sites due to the tumor microenvironment, which is enriched in glucocorticoids and, therefore, impairs antigen presentation and enhances immunosuppression [94].

In conclusion, the mechanisms of resistance to immune checkpoint inhibitors in MSI-H CRC are complex and multifactorial. A better understanding of these mechanisms is essential to develop effective therapeutic strategies to overcome resistance and improve the outcomes of patients with MSI-H CRC.

## 5. Conclusions and Future Directions

Immune checkpoint inhibitors have fundamentally reshaped the prognosis of MSI CRC patients. Despite the low prevalence of MSI in the metastatic setting, representing approximately 5% of these patients, the antitumor effect of such strategies is undeniable. Indeed, around 10% of patients with mCRC will achieve a complete response, almost 50% of whom will have long-lasting responses, highlighting the impressive clinical activity of immune checkpoint inhibitors in this specific patient subgroup. Data from the Keynote-164, CheckMate-142, and Keynote-177 trials suggest that the sooner a patient receives immunotherapy, the better the outcomes. In this regard, the phase III Keynote-177 trial has demonstrated outstanding activity in the first-line setting with anti-PD1 therapy compared to standard chemotherapy, with pembrolizumab doubling the PFS compared to chemotherapy, resulting in it currently being the standard of care. Furthermore, some trials suggest that response rates may improve with longer follow-up periods, as many cases of stabilization eventually transition to response.

Unlike what has been observed in other tumor types, in which there are defined predictive biomarkers of response and well-characterized mechanisms of resistance, biomarkers in the CRC context require further elucidation. Evidence from clinical trials and retrospective cohorts suggests that *RAS* and *BRAF* mutations do not hamper the clinical activity of immunotherapy [95]. Moreover, although the FDA has approved pembrolizumab based on a cutoff > 10 Mut/Mb, CRC non-MSI tumors with high TMB do not achieve clinical benefit, and among MSI patients a well-defined cutoff has not yet been established. Similarly, *B2M* and *JAK1*/2 genomic alterations, which have been identified as mechanisms of resistance in other tumors, are not associated with the clinical outcomes in CRC. Moreover, preclinical evidence suggests that WNT/β-catenin alterations may lead to an immunosuppressive tumor environment, which can enhance immunoregulation. However, there is a lack of clinical evidence in this setting, and it should be also taken into consideration that the WNT pathway is often mutated in CRC. Further clinical validation is needed to confirm these findings. However, one of the most promising biomarkers is the presence of liver metastases. Indeed, the immunosuppressive environment of liver metastases, mostly due to myeloid-derived suppressor cells, macrophages, and an upregulation of TGF-β, leads to a decrease in the number and activity of T cells. Even in real-world cohorts, liver metastases have been correlated with poor response and poor clinical outcomes among MSI tumors treated with immunotherapy [88,89].

In light of these findings, the potential benefit of immunotherapy for MSI mCRC patients is undeniable; however, it remains complicated to identify well-established contraindications to treatment with immune checkpoint inhibitors. For patients with autoimmune disease or solid-organ transplantation, immune checkpoint inhibitors should be considered and discussed with the patient [96,97], as clinical activity has been widely reported. Indeed, despite the fact that approximately 20–30% of patients will develop grade 3 or higher autoimmune-related adverse events, and that there is a risk of organ rejection or a flare of preexisting autoimmune disease, the overall toxicity profile has been proven to be safe and manageable [23,26,98].

Furthermore, even patients with potentially negative biomarkers such as *WNT* mutations or liver metastases can achieve benefit with immunotherapy strategies.

What is clear is that further research is needed to better understand acquired resistance, in order to develop new strategies to overcome such resistance and improve patient outcomes. However, several questions remain unanswered—for example, whether the combination of nivolumab plus ipilimumab will be superior to nivolumab monotherapy as upfront therapy. The CheckMate-8HW study (NCT04008030) is currently comparing nivolumab–ipilimumab vs. nivolumab vs. chemotherapy in the first-line setting. Furthermore, the characteristics of non-responsive patients, representing around 20% of the patients, need to be better elucidated. Indeed, data from CheckMate-142 demonstrated that among non-responders there were several patients with an incorrect MSI diagnosis [26]. The 3–5% disagreement between immunohistochemistry and PCR should be taken into account. Similarly, hyperprogression and pseudoprogression have also been described, particularly under anti-PD1 monotherapy, suggesting that using iRECIST instead of conventional RECIST may help to better identify responders [28,99]. Finally, patients who not respond to immunotherapy may need a different treatment, such as chemotherapy. The phase III COMMIT trial (NCT02997228) is evaluating the anti-PD-L1 agent atezolizumab, alone or in combination with FOLFOX plus bevacizumab, versus standard FOLFOX plus bevacizumab.

Based on the outstanding clinical activity seen with immunotherapy in MSI mCRC, and considering the higher prevalence of MSI in early-stage disease compared to the metastatic scenario, the evaluation of immune checkpoint inhibitors was expanded to earlier disease stages. While trials evaluating immunotherapy in the adjuvant setting are currently recruiting, things have moved fast in the neoadjuvant scenario. Several phase II studies have demonstrated the deep biological impact of using immune checkpoint inhibitors as a neoadjuvant strategy. Based on published data, after immune-based neoadjuvant treatments, complete pathological response was seen in more than 60% of patients and—equally as important—relapses were uncommon. Nevertheless, there are several considerations in this regard. First, the current data are based on phase II, single-arm, small trials with short follow-up periods. Rectal tumor follow-up is easily accessible using rectal examination, MRI, CT scan, or even PET-CT. However, a well-established protocol is needed for the follow-up of distal colon cancer. In this scenario, liquid biopsy may be of help to identify patients at high risk of relapse or with molecular residual disease after the surgery, as this has been proven to be a reliable tool to forecast prognosis and identify minimal residual disease [100,101,102]. Despite the impressive results, the long-term benefit of this strategy and the final impact on OS remain unclear. Another unanswered question is the need for large, randomized, phase III trials to validate these findings.

In conclusion, immune checkpoint inhibitors are highly effective therapeutics for patients with all-stage dMMR/MSI-H CRC, and recent evidence suggests that they are highly promising for neoadjuvant and adjuvant therapies for this patient population. More studies are needed to better define the duration of therapy and whether monotherapy or doublet approaches are needed to achieve or consolidate definitive therapies. Future studies may also reveal the exact role of neoadjuvant versus adjuvant approaches with immune checkpoint inhibitors in patients with dMMR/MSI-H colon cancer, while recognizing that the neoadjuvant strategy should be the preferred strategy in rectal cancer because of the importance of organ preservation.

## Figures and Tables

**Figure 1 cancers-15-04245-f001:**
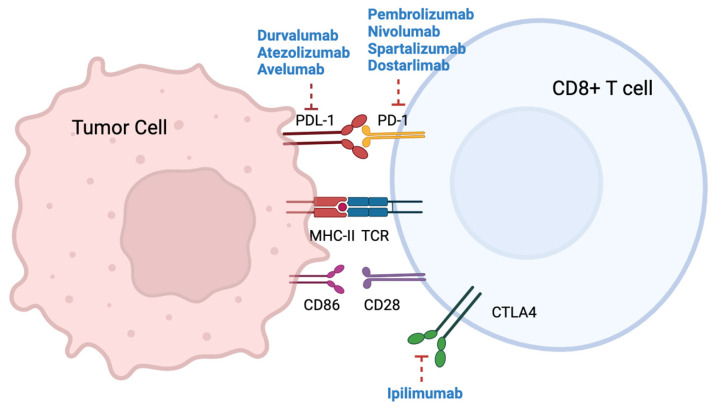
Mechanism of action of immune checkpoint inhibitors.

**Figure 2 cancers-15-04245-f002:**
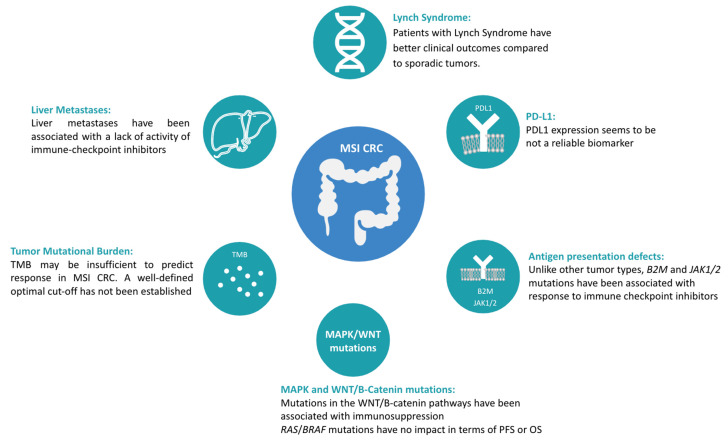
Potential predictive biomarkers evaluated in MSI CRC.

**Table 1 cancers-15-04245-t001:** Summary of completed clinical trials evaluating immune checkpoint inhibitors in MSI metastatic colorectal cancer.

Study	Drug	Sample Size	Phase	Setting	Lynch Syndrome	Right-Sided	Liver Mets	MSI Technique	ORR (%)	CR (%)	PR (%)	SD (%)	PD (%)	Median PFS (mo)	Median OS (mo)	Follow-Up (mo)	Reference
Keynote-164 NCT02460198	Pembrolizumab	61	II	Cohort A: ≥2 prior lines	NA	NA	NA	PCR/IHC	32,8	4.9	27.9	18	45.9	2.3	31.4	62.2	[22]
		63	II	Cohort B: ≥1 prior line	NA	NA	NA	PCR/IHC	34.9	14.3	20.6	20.6	39.7	4.1	47	54.4	[22]
Keynote-177 NCT02563002	Pembrolizumab	153	III	1st line	NA	67%	NA	PCR/IHC	43.8	11.1	32.7	20.9	29.4	16.5	NR	32.4	[23]
Keynote-016 NCT01876511	Pembrolizumab	11	II	≥2 prior lines	82%	NA	55%	PCR	40	0	40	50	10	NR	NR	9	[24]
CheckMate-142 NCT02060188	Nivolumab	74	II	≥1 prior line	38%	NA	NA	PCR/IHC	34	9	24	31	30	6.6	NR	21	[25]
	Nivolumab–ipilimumab	119	II	≥1 prior line	30%	55%	NA	PCR/IHC	65	13	52	21	12	NR	NR	50.9	[26]
	Nivolumab–ipilimumab	45	II	1st line	18%	58%	NA	PCR/IHC	69	13	56	16	13	NR	NR	29	[27]
NIPICOL NCT03350126	Nivolumab–ipilimumab	57	II	≥2 prior lines	67%	54%	NA	NR	NA	NA	NA	NA	NA	3-y 70%	3-y 73%	34.5	[28]
GARNET NCT02715284	Dostarlimab	69	I	≥1 prior line	NR	NR	NA	NR	36.2	2.9	33.3	24.6	30.4	NA	NA	NA	[29]
NCT0315-0706	Avelumab	33	II	≥1 prior line	NA	66.7%	45.5%	PCR/IHC/NGS	24.2	12.1	12.1	54.5	18.2	3.9	13.2	16.3	[30]
SAMCO-PRODIGE 54 NCT03186326	Avelumab	132	II	2nd line	NA	87%	NA	NA	29.5	6.5	23	41	28	3.9	NA	33.3	[31]
NCT02227667	Durvalumab	36	I/II	≥1 prior line	NA	NA	NA	PCR/IHC	22.2	0	8	17	22.2	5.5	NR, 12 mo OS 63.3%	29.16	[32]

ORR, objective response rate; CR, complete response; PR, partial response; SD, stable disease; PD, progressive disease; IHC, immunohistochemistry; Mo, months; Mts, metastases; NA, not available; NR, not reached; y, year.

**Table 2 cancers-15-04245-t002:** Summary of clinical trials evaluating immune checkpoint inhibitors as neoadjuvant therapy in MSI colorectal cancer.

Study	Drug	Phase	Sample Size	Median FU (Months)	Sidedness	Lynch Syndrome	T4/N+	pCR ^$^	MPR ^$$^	Relapse	Reference
NICHE NCT03026140	Nivolumab–ipilimumab	II	21	9 (5.3–15.7)	Left 24%	33	38%/81%	65%	95%	No	[52]
NICHE-2 EudraCT 016-002940-17	Nivolumab–ipilimumab then nivolumab	II	112	13.1 (1.4–57.4)	Left 17%	31%	53%/88%	67%	95%	No	[53]
PICC NCT03926338	Toripalimab	II	17	14.9 (8.8–17)	Left 37%	24% *	74%/84%	88%	94%	No	[54]
	Toripalimab–celecoxib		17		Left 30%	6% *	94%/95%	65%	100%	No	[54]
MSKCC NCT04165772	Dostarlimab	II	12	12 (6–25)	Rectum 100%	57%	19%/94%	100% (cCR)		No	[55]
MDACC NCT04082572	Pembrolizumab	II	27	9.5 (0–26)	NA	37%	61%/79%	79% **	NA	14% ***	[56]

FU: Follow-up, pCR: pathological complete response, MPR: major pathological response, cCR: clinical complete response. ^$^ Pathological complete response: tumors without any viable tumor cells in the resected primary tumor sample and all sampled regional lymph nodes. ^$$^ Major pathological response: presence of 10% or fewer viable tumor cells in the primary tumor. * Patients were considered to have suspected Lynch syndrome if they met the Amsterdam II criteria. ** Pathological response was available in 14 patients. *** Progression events included clinical, radiological, and ctDNA progression.

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
