# Peer review of "Immunotherapy for Colorectal Cancer with High Microsatellite Instability: The Ongoing Search for Biomarkers"

_cancers, 2023, doi:10.3390/cancers15174245_

Round 1

Reviewer 1 Report

CRC with high microsatellite instability usually shows good response to immunotherapy. However, reliable biomarkers are needed in the clinical practice.

The authors should focus on the safety profile of immunotherapy, with particular regard to the risk of reactivation of underlying immune disease (in this regard cite the recent MA: PMID: 33314269)

I would highlight in the title that this paper is a narrative review

Some figures on the mechanism of action of these drugs would be useful.

Author Response

Dear Reviewer, thank you for your comments, 

Reviewer 1: 
The authors should focus on the safety profile of immunotherapy, with particular regard to the risk of reactivation of underlying immune disease (in this regard cite the recent MA: PMID: 33314269). Thank you for your comment. We have included 3-4 lines about the safety profile of ICIs and the aforementioned paper in the references. 

I would highlight in the title that this paper is a narrative review. Thank you for the comment. We have discussed the title with all the authors and, if possible, we prefer the maintain the current title. 

Some figures on the mechanism of action of these drugs would be useful. Thank you for your suggestion. We have included a new Figure that summarizes ICIs' mechanism of action (Figure 1)

Reviewer 2 Report

This review is a great compilation of what is known about immune treatments in colorectal cancer, clinical outcomes and research in the identification of biomarkers. The review is excellent, very complete and instructive. Just minor changes should be addressed: 

Minor changes: 

Table1 and text in section 2.1:

In Table1: Keynote-016 NCT02460198 phase II clinical trial, is incorrect, please change to Keynote-016 NCT01876511

Please add to Table 1 legend: ORR, objective response rate; CR, Complete response; PR, Partial Response; SD, Stable Disease; PD, Progressive Disease

The text in section 2.1: “in patients with refractory colorectal tumors classified as either MSI (11) or MSS (21), as well as a subgroup (which subgroup?) with non-colorectal MSI-positive tumors” (meaning MSS?). This paragraph should be clarified 11 patients are MSI and 21 patients are MSS.

Author Response

Reviewer 2:

Table1 and text in section 2.1: In Table1: Keynote-016 NCT02460198 phase II clinical trial, is incorrect, please change to Keynote-016 NCT01876511. Thank you for your comment. We have included the correct NCT number, thank you.

Please add to Table 1 legend: ORR, objective response rate; CR, Complete response; PR, Partial Response; SD, Stable Disease; PD, Progressive Disease. Thank you, all the abbreviations have been included under table 1. 

The text in section 2.1: “in patients with refractory colorectal tumors classified as either MSI (11) or MSS (21), as well as a subgroup (which subgroup?) with non-colorectal MSI-positive tumors” (meaning MSS?). This paragraph should be clarified 11 patients are MSI and 21 patients are MSS. Thank you for your comment. We have modified the sentence for a better understanding and we also have included the number of MSI mCRC and MSS mCRC included in the trial. 

Round 2

Reviewer 1 Report

The revised manuscript is OK. Thank you!